# Aerobic Capacity in Relation to Selected Elements of Body Posture

**DOI:** 10.3390/ijerph20020903

**Published:** 2023-01-04

**Authors:** Dawid Konrad Mucha, Tomasz Pałka, Renata Skalska-Izdebska, Aneta Teległów, Teresa Mucha, Robert Makuch, Dariusz Mucha

**Affiliations:** 1Department of Medicine and Health Sciences, Andrzej Frycz Modrzewski Krakow University, 30-705 Kraków, Poland; 2Department of Physiology and Biochemistry, Faculty of Physical Education and Sport, University of Physical Education in Kraków, 31-571 Krakow, Poland; 3Department of Physiotherapy, Institute of Health Sciences, Medical College of Rzeszow University, 35-205 Rzeszów, Poland; 4Department of Health Promotion, Institute of Basic Sciences, University of Physical Education in Krakow, 31-571 Krakow, Poland; 5Institute of Health Sciences, Podhale State College of Applied Sciences in Nowy Targ, 34-400 Nowy Targ, Poland; 6Department of Physical Education, Kazimierz Pulaski University of Technology and Humanities in Radom, 26-600 Radom, Poland; 7Department of Body Renovation and Body Posture Correction, Faculty of Physical Education and Sport, University of Physical Education in Kraków, 31-571 Krakow, Poland

**Keywords:** faulty postures, postural defects, aerobic capacity, physical capacity

## Abstract

Background: In the 21st century, with the rapid development of many human life areas, physical activity should be prevalent in health maintenance and promotion. Body posture is a motor habit characteristic of every individual. Its correctness depends on numerous aspects, e.g., physical activity, age, mental state, or eating habits. There are numerous reports in the literature on the impact of physical activity on body posture, correct foot arch development, and the level of aerobic capacity in children and adolescents, but there is a noticeable lack of assessments of these characteristics and their correlations in adults. Aim: To evaluate aerobic capacity in males and females in relation to selected body posture elements. Methods: The study involved 45 females and 46 males aged 20–21 years. The inclusion criteria involved declared good health and no contraindications. Selected somatic traits, body posture, and physical capacity indicators were determined. Results: Physical capacity shows a significant relationship with body mass in both sexes (female: *r* = −0.346; *p* = 0.020; male: *r* = −0.321; *p* = 0.030). A significant correlation was observed between aerobic capacity and lean body mass in females (*r* = −0.428; *p* = 0.003) and body mass (*r* = −0.461; *p* = 0.001) and body fat percentage in males (*r* = −0.443; *p* = 0.002). A significant correlation was demonstrated between maximal oxygen uptake and Clarke’s angle (*r* = −0.300; *p* = 0.045) in females, between maximal oxygen uptake and the loaded area of the right foot (*r* = −0.247; *p* = 0.098) in the male group, and between maximal oxygen uptake and spine lateral deviation (*r* = 0.352; *p* = 0.018) in females. There was no dimorphism between body posture elements and physical capacity except for the level of foot longitudinal arches, feet loading surface, spine lateral deviation, and the range of spine mobility in the sagittal and frontal planes. Aerobic capacity significantly influenced lean body mass (β = −0.379; *p* = 0.007) and spine deviation from the anatomical axis in the frontal plane in females (β = 0.287; *p* = 0.039) and body fat percentage in males (β = −0.443; *p* = 0.002). Conclusions: Selected body posture elements demonstrate relationships with physical capacity in both sexes. The results should find wide practical applications, e.g., in promoting a comprehensive assessment of body posture and physical capacity as determinants of health maintenance.

## 1. Introduction

When taking care of posture, attention should be paid to the physiological ranges of motion in the joints, which are closely linked to mobility and stability, providing an adequate basis for physical and occupational activity. In diagnosing the physical capacity of active people, it is necessary to take into account the level of its aerobic component. The value of maximal oxygen uptake (VO_2_max) in relative terms (mL∙kg^−1^∙min^−1^) is a recognized measure of the body’s aerobic capacity, which depends on the efficiency of organs, the mechanisms involved in tissue oxygen supply, body mass, and the effectiveness of mechanisms participating in energetic processes. In a study conducted among students at the Medical University of Wrocław by Markiewicz–Górka et al. [1], it was shown that only 36.5% of respondents practiced regular physical activity. With the increase in the degree of obesity, a decrease in exercise tolerance is observed [2]. Numerous authors evaluating young people indicate that the male population is more physically active than the female population [3,4]; however, the level of physical activity decreases with age [5]. Similar findings were published by researchers evaluating students [6,7,8,9,10]: a deficit of movement was revealed in any form, which directly reduces the quality of life while impairing fitness and aerobic capacity, whose levels before the age of 25 translate into subsequent periods of ontogenesis [11,12].

Aerobic capacity assessment can predict the body’s response to physical effort. This evaluation can be applied in cardiorespiratory fitness assessment, in competitive sports, and in rehabilitation, although exercise tolerance assessments may be of greater importance in rehabilitation.

There are numerous reports in the literature on the impact of aerobic capacity on body posture, correct foot arch development, and the level of aerobic capacity in children and adolescents, but there is a noticeable lack of assessments of these characteristics and their correlations in adults. The relatively small amount of research undertaken in recent years in the area of body posture and physical capacity has become a rationale for the study of these issues.

The available studies did not look for correlations in the discussed scope, which, according to the authors, is a limitation of the existing research and was the reason for undertaking such an attempt. The aim of this study was to investigate the relationship between aerobic capacity in males and females and selected body posture indicators. The following research questions were posed:Does the level of physical capacity exhibit a relationship with selected body build and posture indicators?Is there any dimorphism in the above relationships?Is the occurrence of the analyzed body posture defects the same in subjects with high and low aerobic capacity?What practical applications can the investigated relationships have?

## 2. Materials and Methods

### 2.1. Study Group

The study involved 91 subjects: 45 females and 46 males aged 20–21 years. It was performed at the Podhale State College of Applied Sciences in Nowy Targ, Poland. The subjects were informed about the study design and provided their written consent to participate, with the possibility to withdraw at any stage without giving any reason. The research was supervised by a paramedic. The inclusion criteria involved declared good health and no contraindications to aerobic capacity tests. The qualifications were carried out by a sports medicine doctor. The individuals entered the tests in sportswear, in the morning hours, at least 2 h after a light meal. The research purpose and essence were explained. The participants were instructed on the particular test execution immediately before their commencement. The administrators of personal data are the authors of this thesis, and the research is stored on the university’s servers. The baseline data obtained are randomized.

### 2.2. Methods

Selected somatic traits, body posture, and aerobic capacity indicators were determined. The first stage of the study comprised the measurement of arterial blood pressure and basic body build indicators: body height, determined with a Martin anthropometer (GPM, Rudolf Martin, LLC, USA) with 0.5 cm accuracy; body weight, assessed with Sartorius F150S-DZA scales (Germany); and body mass index. Body composition was estimated with the 8-electrode bioelectrical impedance method, in accordance with the methodology specified by the device manufacturer (Jawon Medical, certificate CE0197, Korea) and the criteria by Hattori et al. [13]: the investigated indicators were body fat percentage, body fat mass, and lean body mass [14,15]. Following the guidelines [14], the subjects came from one ethnic group, did not suffer from any diseases affecting water–electrolyte balance, did not exert physical effort 2–3 h before the bioelectrical impedance analysis, and had their fluid and food intake times controlled.

For the assessment of body height, body mass, body mass index, and the results in the feet area, comparisons were made between our outcomes and those obtained by Lizis [16]. Body mass index analysis employed the World Health Organization classification [17].

The assessment of foot arches and load distribution was performed with a FreeMed Posture set, consisting of a computer podoscope (2D scanner) and a tensometric mat. Plantocontourogram area values were calculated with an accuracy of 1.0 mm^2^ [18]. The following indicators were used: right and left foot surface (cm^2^), right and left forefoot and hindfoot surface (cm^2^), right and left foot load (%), and right and left forefoot and hindfoot load (%) [18]. From the indicators assessing foot structure, the following values were calculated with a computer program: right and left foot length and width (mm), the surface of the soles of the feet (cm^2^), load distribution on the sole side (cm^2^), longitudinal arches of the right and left foot with Clarke’s angle (°), and transverse arches of the right and left foot determined by the Wejsflog W indicator [18]. The spine and body posture analysis employed the DIERS Formetric III 4D (Diers International GmbH, Germany), and the MediMouse systems (Swiss). The former allowed evaluation of the pelvic tilt (mm), pelvic torsion (°), thoracic kyphosis angle (°), lumbar lordosis angle (°), lateral deviation VPDM (mm) (root mean square, rms), and back surface rotation (mm) (rms). The latter served to determine spine deviation from the anatomical axis in the frontal plane (°), range of spinal mobility in the frontal plane (°), and range of spinal mobility in the sagittal plane in the tested positions: relaxed position, forward bend (°); relaxed position, backward bend (°); and forward bend, backward bend (°).

Physical capacity, often identified with aerobic capacity [3,19,20,21], was assessed with a graded treadmill test until refusal [22,23,24], performed at an ambient temperature of 21.0 ± 0.5 °C and a relative humidity of 40.0 ± 3.0%. The test was preceded by a 3 min warm-up on a cycle ergometer at a pedaling frequency of 60 revolutions∙min^–1^; in males, the intensity equaled 110 W, with power increased by 25 W every 2 min, and in females, the intensity equaled 90 W, with power increased by 20 W every 2 min. The effort continued until the subjective feeling of inability to maintain the desired pedaling rhythm, i.e., until refusal to continue the effort at the specified pedaling resistance. Indicators of exercise respiratory exchange in the graded test were analyzed in 30 s sequences by using a computerized gas analyzer (Cortex, Germany), equipped with oxygen (O_2_) and carbon dioxide (CO_2_) analyzers. Heart rate during the laboratory tests was telemetrically recorded with a Polar M400 heart rate monitor (Polar Electro, Finland).

### 2.3. Statistical Analysis

The results were processed with Statistica software, ver. 13 (StatSoft, Tulsa, OK, USA). The Shapiro–Wilk test determined whether the variables in each group approximated a normal distribution. Correlations between variables were investigated with two correlation tests: Pearson’s *r* correlation test and Spearman’s rho correlation test. The test choice depended on whether the examined variables had a near-normal distribution and whether they were ordinal or ratio variables. When the distribution was close to normal and the variable was a ratio variable, Pearson’s *r* correlation test was used; otherwise, Spearman’s rho correlation test was applied. A stepwise multiple regression analysis determined the set of investigated characteristics that influenced aerobic capacity in males and females. In all analyses, effects for which the probability value *p* was less than the adopted significance level of α = 0.05 were assumed significant (*p* < 0.05).

## 3. Results

First, descriptive statistics for the subjects’ basic somatic traits and aerobic capacity are presented. Then, we provide the results of correlation analyses aiming to determine the relationships between aerobic capacity and selected body posture indicators. The final stage is to present the results of the stepwise multiple regression analysis, which established the relationships between selected body posture elements and the aerobic capacity level of the examined males and females.

Mean body height was 166.6 ± 5.70 cm in females and 179.0 ± 5.50 cm in males. As for body mass, the mean values equaled 61.6 ± 8.60 kg in females and 80.5 ± 10.90 kg in males (Table 1).

The mean value of VO_2_max was 30.80 ± 4.70 mL∙kg^−1^∙min^−1^ in females and 35.6 ± 7.80 mL kg^−1^ min^−1^ in males (Table 2).

The obtained values were transformed in accordance with the American Heart Association standards [25]. For females aged 20–29 years, a low level of VO_2_max is considered to be below 24.00 mL∙kg^−1^∙min^−1^, a sufficient level is 24.00–30.00 mL∙kg^−1^∙min^−1^, and an average level is in the range of 31.00–37.00 mL∙kg^−1^∙min^−1^. An indicator value ranging 38.00–48.00 mL∙kg^−1^ min^−1^ implies a good VO_2_max level. Any value that is 49.00 mL∙kg^−1^∙min^−1^ or more is regarded as high VO_2_max under conditions of maximum effort. For males aged 20–29 years, a low level of VO_2_max is considered to be below 25.00 mL∙kg^−1^∙min^−1^, a sufficient level is 25.00–33.00 mL∙kg^−1^∙min^−1^, and an average level is in the range of 34.00–42.00 mL∙kg^−1^∙min^−1^. An indicator value ranging 43.00–52.00 mL∙kg^−1^∙min^−1^ implies a good VO_2_max level. Any value that is 53.00 mL∙kg^−1^∙min^−1^ or more is regarded as high aerobic capacity.

With regard to VO_2_max, a low level was recorded in 53.33% of females and 43.48% of males. Average aerobic capacity was indicated in 35.56% of females and 45.65% of males. Only 11.11% of females and 10.87% of males were characterized by a good level of VO_2_max. A sufficient or high VO_2_max level (by the American Heart Association standards [25]) was not achieved by any of the participants.

Both males and females presented a statistically significant negative correlation between VO_2_max and body mass (female: *r* = −0.346; *p* = 0.020; male: *r* = −0.321; *p* = 0.030). Moreover, in the female group, a statistically significant negative correlation was observed between lean body mass and VO_2_max (*r* = −0.428; *p* = 0.003). Among males, there were statistically significant negative correlations between aerobic capacity and body fat mass (*r* = −0.461; *p* = 0.001) as well as the indicator value converted to % of body mass (*r* = −0.443; *p* = 0.002). A similar correlation (but high in this case) was found between VO_2_max values and body fat mass (*r* = −0.531; *p* > 0.001) and body fat percentage (*r* = −0.613; *p* > 0.001). These results indicate that males with a higher body fat percentage exhibited higher aerobic capacity values (Table 3).

Among females, a statistically significant negative correlation was observed between VO_2_max and right foot Clarke’s angle (*r* = −0.300; *p* = 0.045). There was also a trend (0.1 > *p* > 0.05) implying a negative weak relationship between right foot longitudinal arches and VO_2_max (*r* = −0.276; *p* = 0.066) (Table 4).

In the male group, a trend was noted (0.1 > *p* > 0.05) implying a negative weak relationship between left foot percentage load and VO_2_max (*r* = −0.247; *p* = 0.098). There was also a trend (0.1 > *p* > 0.05) indicating a positive weak relationship between right foot percentage load and VO_2_max (*r* = 0.247; *p* = 0.098) (Table 5).

For the analyses of the correlation of pelvic tilt and pelvic torsion with aerobic capacity, the nonparametric Spearman’s rho correlation test was used owing to the smaller number of subjects (resulting from the deviations to the left or right side). The results did not indicate statistically significant relationships between these variables (*p* > 0.05). In the female group, there were trends (0.10 > *p* > 0.05) suggesting a relationship between VO_2_max and pelvic tilt to the left (*r*_s_ = 0.360; *p* = 0.065) and pelvic torsion to the left (*r*_s_ = −0.432; *p* = 0.083). Furthermore, in the female group, Pearson’s *r* correlation test revealed a statistically significant positive correlation between VO_2_max and lateral deviation VPDM (*r* = 0.352; *p* = 0.018) (Table 6).

Among males, the analysis with Pearson’s *r* correlation test indicated a statistically significant positive weak correlation of VO_2_max with the left-side spinal range of mobility (*r* = 0.292; *p* = 0.049). In the male group, a trend was also observed (0.1 > *p* > 0.05) implying positive weak relationships between VO_2_max and left-side (*r* = 0.259; *p* = 0.082) and right-side (*r* = 0.249; *p* = 0.095) mobility, total range of mobility in the frontal plane (*r* = 0.265; *p* = 0.075), and range of mobility in backward bend (*r* = 0.247; *p* = 0.098) (Table 7).

Two variables are significant in the regression equation; in the female group, VO_2_max was influenced by lean body mass (β = −0.379; *p* = 0.007) and lateral deviation VPDM (β = 0.287; *p* = 0.039). The β coefficient is negative and positive for these two variables, which, with the assumption of linear dependence, would indicate an inversely proportional relationship in the case of lean body mass and a directly proportional relationship in the case of lateral deviation VPDM. The multiple determination coefficient equaled 0.228 (model 2), which implies that ca. 23% of the variation in VO_2_max can be explained on the basis of these variables. This variation is best described by lean body mass (*R*^2^ = 0.164), followed by lateral deviation VPDM (model 2–1 difference: *R*^2^ = 0.123) (Table 8).

The variable analysis in the regression equation revealed that VO_2_max was influenced by lean body mass (β = −0.379; *p* = 0.007) and spine deviation from the anatomical axis in the frontal plane (β = 0.287; *p* = 0.039) in females and by body fat percentage (β = −0.443; *p* = 0.002) in males (Table 9).

## 4. Discussion

The integration of the particular systems of the human body is fundamental to its functioning. In recent decades, the postural control system has been the subject of intense research, but there are few reports on the relationships between body posture and physical capacity levels.

Analyzing the basic somatic traits in the examined males and females, one may notice that the obtained values are at a level similar to those achieved by other researchers [26,27,28,29,30].

Lavie et al. [31] observed that the relationships between body mass index and chronological age on the one hand and physical capacity on the other were inversely proportional. It is worth noting that overweight, obesity, and physical inactivity associated with low levels of capacity are factors influencing mortality. It has been shown that the training level and aerobic capacity in overweight individuals are generally lower than among people with a normal body mass index [32]. Numerous authors evaluating young people indicate that the male population is more physically active than the female population [3,4]; however, the level of physical activity decreases with age [5]. Similar findings were published by researchers evaluating students [6,7,8,9,10]: a deficit of movement was revealed in any form, which directly reduces the quality of life while impairing fitness and aerobic capacity, whose levels before the age of 25 translate into subsequent periods of ontogenesis [11,12].

VO_2_max values are highly individualized. It is assumed that the lowest value allowing full locomotor independence is 15 mL∙kg^−1^∙min^−1^ [33]; the lowest levels are found in people with cardiopulmonary failure and in seniors, with healthy individuals ranging 15–85 mL∙kg^−1^∙min^−1^. Tyka [20] points to the important fact that the level of VO_2_max is an inborn trait and susceptible to physical training to a limited extent only (ca. 25%). Shete et al. [34] established VO_2_max among 25 females as 39.62 ± 2.80 mL∙kg^−1^∙min^−1^ in training individuals and as 23.54 ± 3.26 mL∙kg^−1^∙min^−1^ in nontraining individuals, with a mean value of 30.80 mL∙kg^−1^∙min^−1^. Values typical of young, healthy students are within the range of 45.00–55.00 mL∙kg^−1^∙min^−1^ [20], which is not consistent with the results obtained in the present study. In a study performed by Pałka [35] among university students, the highest level of VO_2_max expressed in mL∙kg^−1^∙min^−1^ was reported in students of the University of Physical Education in Krakow, Poland. The mean indicator equaled 55.03 ± 3.82 mL∙kg^−1^∙min^−1^, which may imply that these students undertake much physical activity in different forms in their free time. Not without significance are also the physical activities included in the study curriculum, which positively influence the development and maintenance of high VO_2_max. Pałka [35] recorded the lowest VO_2_max in students of the State University of Applied Sciences in Nowy Sacz, Poland; the mean values equaled 42.89 ± 6.33 mL∙kg^−1^∙min^−1^, which may indicate less regular physical activity in their leisure time, a smaller amount of physical activity at the university, or a weaker aerobic potential. Nabi et al. [36], in their research among 57 students (30 males and 27 females), revealed mean VO_2_max values of 45.66 ± 8.90 mL∙kg^−1^∙min^−1^ in males and 37.85 ± 4.30 mL∙kg^−1^∙min^−1^ in females. The results for students of the State University of Applied Sciences in Nowy Sacz, Poland, were at a significantly higher level than those for both male and female respondents and did not considerably differ from the VO_2_max outcomes obtained in the Polish national ice hockey team (51.6 ± 4.11 mL∙kg^−1^∙min^−1^) [37] or second-league volleyball players (47.61 ± 7.12 mL∙kg^−1^∙min^−1^) [38]. The VO_2_max results achieved by students of the University of Physical Education in Krakow, Poland, are comparable to those reported by Tyka et al. [39] among first-league footballers (56.48 ± 4.80 mL∙kg^−1^∙min^−1^) and snowboarders (55.89 ± 4.17 mL∙kg^−1^∙min^−1^).

When analyzing the relationships between somatic characteristics and physical capacity of the examined subjects, a statistically significant negative correlation was found between VO_2_max and body mass in females (*r* = −0.346; *p* = 0.020) and in males (*r* = −0.321; *p* = 0.030). This implies that people with higher body mass are characterized by higher VO_2_max values. Moreover, in the female group, a statistically significant negative correlation was observed between lean body mass and VO_2_max (*r* = −0.428; *p* = 0.003): the lower the lean body mass, the higher the results on the VO_2_max test.

Shete et al. [34], in their study concerning the relationship between VO_2_max and body fat percentage in training and nontraining females, showed that the training females exhibited higher VO_2_max and noted a correlation, although not statistically significant, between the indicators in question. In the present study, statistically significant negative correlations were also reported between VO_2_max and body fat mass (*r* = −0.461; *p* = 0.001) as well as the indicator value converted to % of body mass (*r* = −0.443; *p* = 0.002). In females, there was additionally a trend (0.1 > *p* > 0.05) suggesting a negative weak relationship between body mass index and VO_2_max (*r* = −0.271; *p* = 0.072).

There is a significant discrepancy in the literature regarding the occurrence of postural defects. This refers to the area of feet, lower limbs, position of the pelvis and spine, or defects within the trunk. The inconsistency of the presented results is due to different research methodologies, e.g., in assessing spinal mobility, and also to limitations in equipment [40]; reference values for students aged 19–25 years are also sometimes missing. The academic community, owing to their lifestyles, courses of study, and the lack of broad diagnostics, are becoming a risk group for postural defects, reduced fitness levels, or threats arising from sudden attempts to change their attitude towards physical activity (lack of knowledge, diagnostics, and individualized training plan adjusted to physical fitness and capacity). In spite of considerable technological progress, test results obtained with a given device are still not properly interpreted, which results, among others, from the lack of standards defined in the literature. An example is the foot longitudinal arch indicator, determined with Clarke’s angle. In this case, the researcher is at risk of misinterpreting the results because Clarke’s angle is determined manually rather than automatically by an algorithm, as with DIERS, which is an optoelectronic system that automatically recalculates the obtained measurement values, and the investigator’s only task is to follow the algorithm provided by the manufacturer.

In terms of foot arches, in the female group, there was a trend (0.1 > *p* > 0.05) implying a negative weak relationship between the right foot longitudinal arch indicator and VO_2_max (*r* = −0.276; *p* = 0.066). In the male group, trends were observed (0.1 > *p* > 0.05) suggesting a negative weak relationship between left foot percentage load and VO_2_max (*r* = −0.247; *p* = 0.098) and a positive weak relationship between right foot sole percentage load and VO_2_max (*r* = 0.247; *p* = 0.098).

As for pelvic and spinal position, in the female group, a statistically significant correlation was revealed between VO_2_max and lateral spine deviation VPDM (*r* = 0.352; *p* = 0.018). Moreover, there were trends (0.10 > *p* > 0.05) indicating a relationship between VO_2_max and back surface rotation (r = 0.275; *p* = 0.067), as well as pelvic tilt to the left (rs = 0.360; *p* = 0.065) and pelvic torsion to the left (rs = −0.432; *p* = 0.083).

In turn, in the male group, trends were reported (0.1 > *p* > 0.05) implying positive weak relationships between VO2max and left-side (r = 0.259; *p* = 0.082), right-side (r = 0.249; *p* = 0.095), and total spine mobility in the frontal plane (r = 0.265; *p* = 0.075), as well as backward bend range of spine mobility (r = 0.247; *p* = 0.098). The presented group was weak in terms of VO2_2_max according to American Heart Association standards [25]. In terms of limitations, it should be pointed out that the group size is too small and its environmental diversity is lacking in the context of generalizing conclusions. There is no possibility to compare the research results with similar ones found in the professional literature, which prompts the authors to continue the research on a larger population.

## 5. Conclusions

Detailed analysis in the area of the selected elements of body posture and the level of aerobic capacity in adults supplement the published assessments and correlations in this area. The obtained results should find wide practical applications, e.g., in promoting a comprehensive assessment of body posture and physical capacity as determinants of health maintenance.

The level of aerobic capacity shows a statistically significant relationship with body mass in both sexes. A statistically significant correlation was observed between aerobic capacity and lean body mass in females and body mass and body fat percentage in males. Among females, a statistically significant correlation was demonstrated between VO_2_max and Clarke’s angle, the loaded area of the right foot, and spine lateral deviation.

There was no dimorphism in the investigated relationships between the selected body posture elements and physical capacity except for the level of foot longitudinal arches, feet loading surface, lateral deviation of the spine, and the range of its mobility in the sagittal and frontal planes.

Aerobic capacity level significantly determined lean body mass and spine deviation from the anatomical axis in the frontal plane in females and body fat percentage in males.

Knowing the correlations among the analyzed elements, it will be possible to select the optimal program of physiotherapeutic treatment in practical activities with patients, which will contribute to the improvement of its effectiveness, and above all, the health safety of patients.

## Figures and Tables

**Table 1 ijerph-20-00903-t001:** Descriptive statistics for basic somatic characteristics and body composition.

Variable	Female				Male			
	Minimum	Maximum	Mean	Standard Deviation	Minimum	Maximum	Mean	Standard Deviation
BH (cm)	154.00	180.00	166.60	5.70	170.00	192.00	179.0	5.50
BM (kg)	44.50	78.70	61.70	8.60	53.70	116.20	80.50	10.90
BMI (kg m^−2^)	16.00	25.00	18.80	2.70	15.52	29.58	22.60	2.80
LBM (kg)	38.70	51.20	44.60	4.27	55.60	70.08	62.75	6.28
BFM (kg)	10.25	21.50	16.72	4.72	11.45	24.35	17.09	6.22
BFP (%)	8.70	23.80	26.97	5.27	5.90	29.92	20.89	5.44

BH—body height, BM—body mass, BMI—body mass index, LBM—lean body mass, BFM—body fat mass, BFP—body fat percentage.

**Table 2 ijerph-20-00903-t002:** Descriptive statistics for the subjects’ maximal oxygen uptake.

Variable	Female				Male			
	Minimum	Maximum	Mean	Standard Deviation	Minimum	Maximum	Mean	Standard Deviation
VO_2_max (mL∙kg^−1^ min^−1^)	23.80	42.50	30.80	4.70	21.40	52.40	35.60	7.80

VO_2_max—maximal oxygen uptake.

**Table 3 ijerph-20-00903-t003:** Results of Pearson’s *r* correlation test examining relationships between maximal oxygen uptake and selected somatic traits.

Variable		Female (*n* = 45)	Male (*n* = 46)
		VO_2_max	VO_2_max
BH	*r*	−0.103	−0.112
	*p*	0.499	0.460
BM	*r*	−0.346 *	−0.321 *
	*p*	0.020	0.030
BMI	*r*	−0.271	−0.246
	*p*	0.072	0.100
LBM	*r*	−0.428 **	−0.166
	*p*	0.003	0.269
BFM	*r*	−0.228	−0.461 **
	*p*	0.133	0.001
BFP	*r*	−0.054	−0.443 **
	*p*	0.725	0.002

VO_2_max—maximal oxygen uptake, BH—body height, BM—body mass, BMI—body mass index, LBM—lean body mass, BFM—body fat mass, BFP—body fat percentage. * Correlation significant at the level of 0.05 (two-sided). ** Correlation significant at the level of 0.01 (two-sided). *p*—probability value. *r*—correlation coefficient.

**Table 4 ijerph-20-00903-t004:** Results of Pearson’s *r* correlation test examining relationships between maximal oxygen uptake and feet somatic characteristics.

Variable		Female (*n* = 45)	Male (*n* = 46)
		VO_2_max	VO_2_max
Left foot length	*r*	0.127	0.135
	*p*	0.406	0.371
Right foot length	r	0.137	0.128
	*p*	0.371	0.395
Left foot width	*r*	0.178	0.244
	*p*	0.243	0.102
Right foot width	*r*	0.107	0.201
	*p*	0.485	0.182
Left foot Wejsflog indicator	*r*	−0.040	−0.154
	*p*	0.793	0.308
Right foot Wejsflog indicator	*r*	0.054	−0.079
	*p*	0.723	0.600
Right foot Clarke’s angle	*r*	−0.276	0.231
	*p*	0.066	0.122
Left foot Clarke’s angle	*r*	−0.192	0.218
	*p*	0.206	0.146

VO_2_max—maximal oxygen uptake. *p*—probability value. *r*—correlation coefficient.

**Table 5 ijerph-20-00903-t005:** Results of Pearson’s *r* correlation test examining relationships between maximal oxygen uptake and foot load indicators.

Variable		Female (*n* = 45)	Male (*n* = 46)
		VO_2_max	VO_2_max
Left foot surface	*r*	0.165	−0.232
	*p*	0.278	0.121
Right foot surface	*r*	0.099	−0.132
	*p*	0.519	0.383
Left foot percentage load	*r*	−0.136	−0.247
	*p*	0.373	0.098
Right foot percentage load	*r*	0.136	0.247
	*p*	0.373	0.098

VO_2_max—maximal oxygen uptake. *p*—probability value. *r*—correlation coefficient.

**Table 6 ijerph-20-00903-t006:** Relationships between maximal oxygen uptake and pelvic position, angle of spinal kyphosis and lordosis, and body posture indicators.

Variable		Female (*n* = 45)	Male (*n* = 46)
		VO_2_max	VO_2_max
Pelvic tilt to the right ^a^	*r* _s_	0.169	−0.155
	*p*	0.600	0.597
Pelvic tilt to the left ^a^	*r* _s_	0.360	−0.310
	*p*	0.065	0.150
Pelvic torsion to the right ^a^	*r* _s_	−0.047	0.257
	*p*	0.815	0.188
Pelvic torsion to the left ^a^	*r* _s_	−0.432	−0.367
	*p*	0.083	0.162
Kyphosis angle	*r*	0.016	0.028
	*p*	0.916	0.856
Lordosis angle	*r*	−0.109	−0.094
	*p*	0.477	0.536
Back surface rotation	*r*	0.275	0.019
	*p*	0.067	0.902
Lateral deviation VPDM	*r*	0.352 *	−0.156
	*p*	0.018	0.299

VO_2_max—maximal oxygen uptake. ^a^ Spearman’s rho correlations calculated for subjects with pelvic tilt/torsion. * Correlation significant at the level of 0.05 (two-sided). *p*—probability value. *r_s_*—correlation coefficient.

**Table 7 ijerph-20-00903-t007:** Results of Pearson’s *r* correlation test examining relationships between maximal oxygen uptake and the spinal range of mobility in the sagittal and frontal planes.

Variable		Female (*n* = 45)	Male (*n* = 46)
		VO_2_max	VO_2_max
Frontal plane: left	*r*	0.062	0.292 *
	*p*	0.687	0.049
Frontal plane: extension	*r*	−0.059	0.187
	*p*	0.701	0.212
Frontal plane: right	*r*	0.231	0.213
	*p*	0.126	0.156
Mobility: left extension	*r*	0.080	0.259
	*p*	0.603	0.082
Mobility: right extension	*r*	0.207	0.249
	*p*	0.173	0.095
Mobility: left–right	*r*	0.055	0.265
	*p*	0.721	0.075
Spinal position in the sagittal plane	*r*	−0.009	−0.107
	*p*	0.956	0.477
Sagittal plane: bend	*r*	0.116	0.047
	*p*	0.447	0.755
Sagittal plane: hyperextension (−)	*r*	0.181	0.280
	*p*	0.233	0.059
Mobility: extension-bend	*r*	0.113	0.078
	*p*	0.459	0.605
Mobility: extension-hyperextension (−)	*r*	0.186	0.247
	*p*	0.220	0.098
Mobility: bend-hyperextension	*r*	0.172	0.175
	*p*	0.259	0.245

VO_2_max—maximal oxygen uptake. * Correlation significant at the level of 0.05 (two-sided). *p*—probability value. *r*—correlation coefficient.

**Table 8 ijerph-20-00903-t008:** Stepwise multiple regression of maximal oxygen uptake in relation to lean body mass and lateral deviation VPDM in the female group.

Model	Variable	*B **	Standard Error	β	*t*	*p*
1	Constant	52.042	6.855		7.592	0.000
	Lean body mass	−0.476	0.153	−0.428	−3.108	0.003
	Multiple correlation coefficient: *R* = 0.438 Multiple determination coefficient: *R*^2^ = 0.164 Equation significance: *F* = 9.558; *p* = 0.003					
2	Constant	47.610	6.909		6.891	0.000
	Lean body mass	−0.420	0.149	−0.379	−2.816	0.007
	Lateral deviation VPDM	0.378	0.177	0.287	2.133	0.039
	Multiple correlation coefficient: *R* = 0.513 Multiple determination coefficient: *R*^2^ = 0.228 Equation significance: *F* = 7.502; *p* = 0.002					

* Refers to the unstandardized coefficient. β—standardized regression coefficient. *t*—significance level of β. F—equation significance.

**Table 9 ijerph-20-00903-t009:** Stepwise multiple regression of maximal oxygen uptake in relation to body fat percentage in the male group.

Model	Variable	*B **	Standard error	β	*t*	*p*
1	Constant	48.835	4.154		11.755	>0.001
	Body fat percentage	−0.632	0.193	−0.443	−3.281	0.002
	Multiple correlation coefficient: *R* = 0.443 Multiple determination coefficient: *R*^2^ = 0.178 Equation significance: *F* = 10.783; *p* = 0.002					

* Refers to the unstandardized coefficient. β—standardized regression coefficient. *t*—significance level of β. F—equation significance.

## Data Availability

All data generated or analyzed during the study are included in this article.

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
