# Peer review of "Aerobic Capacity in Relation to Selected Elements of Body Posture"

_ijerph, 2023, doi:10.3390/ijerph20020903_

Round 1
Reviewer 1 Report
Research results from different countries can be given in comparisons.
Author Response
Dear Reviewer,
We would like to thank you for your review
Best regards,
Reviewer 2 Report
Thank you for your submission.
The manuscript reports a result of the relationship between aerobic capacity and certain body posture components. The results showed that select body posture elements (e.g., body mass) demonstrate relationships with physical capacity in both sexes. I fail to understand the importance of this research to progress the current research in the area?
Comments for the Authors
1. Firstly, the authors are recommended to overhaul the introduction and discussion sections. The reader might find it hard to follow the intro as the rationales of the current study is not well-presented. The discussion falls short of showing general coverage of current findings and the authors may want to deepen the relevant evidence to support the results of this study.
2. Secondly, the authors may want to strengthen the theoretical background to build up the research questions. This is essential to explain the current findings in the discussion as well.
3. Please proofread the entire manuscript. For example, why is figure 1 in another language?
Author Response
Dear Reviewer,
We would like to thank you for your review.
1. We add several bibliography position in terms of physiology and body posture. The available studies did not look for correlations in the discussed scope, which, according to the authors, is a limitation of the research and was the reason for undertaking such an attempt. In terms of limitations, it should be pointed out that the group size is too small and its environmental diversity in the context of generalizing conclusions. There is no possibility to compare the research results with similar ones found in the professional literature, which prompts the authors to continue the research on a larger population.
2. The justification for the undertaken research was specified in the introduction, which was also reflected in the final part of the discussion
3. We decided to delate the Figure 1.
Reviewer 3 Report
Introduction
In the introduction, several general statements are made, however there is the lack of references (lines 45-62). In my opinion, the authors should better describe the background of their study.
Line 63 – please, correct the sentences: the study assessed (investigated) … or the aim of this study was to investigate (to assess)…
Material and Methods
Declared good health and no contraindications – it should be explained. No contraindications to what?
How were the respondents selected? Out of how many people? Was the selection for the study random? What is the authors' justification for the age of the subjects 20-21 years old? Was age one of the selection criteria for the study?
Lines 107-108 – the lack of references.
How was the Spearman and Person correlation coefficient qualitatively evaluated? It should be given in statistical analysis section.
Results
Figure 1 is in Polish! Please, correct that. I suggest that in Figure 1 under the specified level of VO2MAX, the values (cut-off points) can be given.
How do the authors explain the result of low VO2MAX levels in more than a half of the women studied? In that case, did VO2MAX have a normal distribution?
In the descriptive statistics (tables 1-2) I suggest to provide confidence intervals (C.I.), minimal and maximal values.
Why didn’t the Authors provide the results of the other parameters (mobility, sagittal spinal curvatures, etc.)?
There is a lack of information on whether overweight or obese people were among those surveyed.
Lines 171-175 repeat the results from Figure 1.
What is “average correlations”?
Please correct “P” on “p” in the text and tables.
Discussion
In the discussion, the authors only compare the results of VO2max levels and refer to studies on physical activity levels. However, they do not evaluate it themselves. Rather, the discussion should have focused on answering the research questions posed and explaining the relationships obtained.
The last two paragraphs of the discussion repeat the results of the study.
Author Response
Dear Reviewer,
We would like to thank you for your review.
Introduction
Corrections have been made throughout the lines 45-62 and also in the line 63 as suggested.
Material and methods
The inclusion criteria involved declared good health and no contraindications to aerobic capacity tests. The qualifications were carried out by a sports medicine doctor.
The choice of respondents was intentional. The only university educating in the field of physiotherapy in Podhale was indicated. The research covered the first and second year of physiotherapy students, 90 were selected from a group of 140.The exclusion criteria were obesity, lack of consent and medical qualifications. In addition to age, the criterion was a doctor's consent and the chosen field of study. In addition, the authors of the study are physiotherapists, which also contributed to undertaking these studies.
Lines 107-108 – the lack of references - thank you for your comment. Reference added.
The correlation between the variables was tested using two correlation tests: Pearson's r-correlation test and Sperman's rho-correlation test. The choice of the correlation test depended on whether the examined variables had a distribution of results close to normal and on the fact whether these variables were ordinal or quotient. If the distribution of results was close to normal and the variable was quotient, the Persona r correlation test was used, otherwise the Sperman rho correlation test.
Results
We decided to delete the Figure 1.
We decided to add ,,Numerous authors evaluating young people indicate that the male population is more physically active than the female population [10, 22]; however, the level of physical activity decreases with age [23]. Similar findings were published by researchers evaluating students [24–28]: a deficit of movement was revealed in any form, which directly reduces quality of life while impairing fitness and aerobic capacity, whose level before the age of 25 translates into subsequent periods of ontogenesis [29, 30]."
The Shapiro-Wilk test was used to check whether the distribution of the examined variables in each group is close to the normal distribution. The study group had a normal distribution.
We provided confidence intervals in tables 1-2 as suggested. The authors did not provide the following indicators, because they had no impact on the formulated questions and conclusions.
Obese people were excluded from the study - other information related to somatic features is presented in the table.
We decided to remove the Figure 1.
Due to average correlations - it was the translator fault – corrected.
Due to ,,P" on ,,p" - all changes have been made.
Discussion
We changed the end of discussion according to your recommendation.
,,The presented group was weak in terms of vo2max according to American Heart Association standards [14]. In terms of limitations, it should be pointed out that the group size is too small and its environmental diversity in the context of generalizing conclusions. There is no possibility to compare the research results with similar ones found in the professional literature, which prompts the authors to continue the research on a larger population"
Reviewer 4 Report
Title: Aerobic capacity in relation to selected elements of body posture in men and women
Manuscript ID: ijerph-2042652
This manuscript presented an interesting study of the relationship between the elements of body posture and aerobic capacity, i.e., the maximum amount of oxygen that a subject can use per unit of time and body weight. The paper is written well, however, it can be strengthened further. Please note the following comments to consider and address:
1. In the title – “Aerobic capacity in relation to selected elements of body posture in men and women” – words men and women can be redundant. If necessary to keep, try using more suitable words males and females.
2. Also, at many places, male and female words are interchangeably used for men and women. Please follow a consistent pattern of words.
3. Page 1, lines 21~23 – “the impact of physical activity on body posture, correct foot arches development, and the level of physical capacity in children and adolescents, but there is a noticeable lack of assessment of these characteristics and their correlations in adults.” – does the term physical activity mean aerobic capacity? Similarly, on page 2, line 57, is the term physical activity used in place of aerobic capacity? Words physical activity, physical capacity, and aerobic capacity are quite mixed-up, it is advised to use ‘clear’ words to avoid any ambiguity in the meaning of the sentences.
4. “1. Introduction” – this section is quite short, discussed very briefly and surprisingly no previous work and their limitations are given, no citations in the introduction. Authors must rewrite the introduction section citing previous related work and their limitations, followed by how this study proposed to overcome these limitations.
5. Methods section is very well written and explained how different devices and systems were used for collecting the data, still it doesn’t discuss anything about the type and nature of the data. Authors may discuss briefly about the data, data curation and processing.
6. Page 4, Figure 1., ‘Polish’ terms used (may be inadvertently) – such as Kobiety is used for female/ women, and mężczyźni is used and for male/men. Please correct it and change it to standard English words. Also, use the decimal point (.) instead of comma (,) to represent real numbers, such as 53,33 should be changed to 53.33 (may be this is due to standard EU practice). Also, label the abscissa and ordinate, and in the bar-chart skip the ‘%’, just write the real number.
7. Page 3, lines 136-137 – “A stepwise multiple regression analysis determined the set of investigated characteristics that influenced aerobic capacity in men and women.” –authors may discuss more about this analysis.
8. Page 7, line 230 – “Two variables are significant in the regression equation.” – which two variables are referred in this sentence?
9. “3. Results” and “4. Discussion”– are written and explained very well, but authors should also discuss about the limitation of this study.
However, this manuscript is written very well and addressed an interesting issue which is usually overlooked, some changes are still required to improve the quality of this paper.
Author Response
Dear Reviewer,
We would like to thank You for Your review.
- The title has been changed as suggested by the reviewer.
-
Corrections have been made throughout the article as suggested
-
Corrections have been made throughout the article as suggested – we mentioned aerobic capacity.
-
We added several bibliography position in terms of physiology and body posture. The available studies did not look for correlations in the discussed scope, which, according to the authors, is a limitation of the research and was the reason for undertaking such an attempt.
-
The administrator of personal data are the authors of the thesis, and the research is stored on the university's servers. The baseline data obtained are randomized.
- We decided to delate the figure.
-
The correlation between the variables was tested using two correlation tests: Pearson's r-correlation test and Sperman's rho-correlation test. The choice of the correlation test depended on whether the examined variables had a distribution of results close to normal and on the fact whether these variables were ordinal or quotient. If the distribution of results was close to normal and the variable was quotient, the Persona r correlation test was used, otherwise the Sperman rho correlation test.
- Two variables are significant in the regression equation, in the female group, VO2max was influenced by lean body mass (β = –0.379; p = 0.007) and lateral deviation VPDM (β = 0.287; p = 0.039).
-
In terms of limitations, it should be pointed out that the group size is too small and its environmental diversity in the context of generalizing conclusions. There is no possibility to compare the research results with similar ones found in the professional literature, which prompts the authors to continue the research on a larger population.
Once again, we would like to thank You for Your review.
Best regards,
Authors
Round 2
Reviewer 4 Report
Title: Aerobic Capacity in Relation to Selected Elements of Body Posture
Manuscript ID: ijerph-2042652
This review is in response to the manuscript revised as per the comments on December 15, 2022. The revised manuscript has addressed and incorporated all comments. Revised version explained well, still there is some possibility for minor corrections, as:
1. ‘men’ and ‘women’ words are used at number of places as well as ‘male’ and ‘female’. I think terms male/ female are age-independent terms, so it should be used.
2. Page 4, Table 1. and Table 2. – long form of MIN, MAX, M, and SD should be mentioned at the bottom of the table to avoid any misunderstanding.
3. Page 5, Table 3. and Table 4.; Page 6, Table 5. and Table 6.; Page 7, Table 7. – upper case ‘P’ is used which is not similar as rest of the text.
4. Page 7, Table 8. – please use the long form of third column title ‘B’, it creates an ambiguity with ‘β’.
Please do take care of above suggestions, I do not have any comments. Also, check for typos carefully, if any.
Good luck.

Author Response
Dear Reviewer,
- Once again thank you for your feedback, we made changes according to your comments.
- We provided long forms of MIN, MAX, M, and SD.
- We changed the upper case.
- In case of ,,B" we added reference below the table ,,*refers to the unstandardized coefficient"
Best regards,
